# Low-Level Laser Therapy for the Treatment of Oral Mucositis Induced by Hematopoietic Stem Cell Transplantation: A Systematic Review with Meta-Analysis

**DOI:** 10.3390/medicina59081413

**Published:** 2023-08-03

**Authors:** Rocco Franco, Ettore Lupi, Enzo Iacomino, Angela Galeotti, Mario Capogreco, João Miguel Marques Santos, Maurizio D’Amario

**Affiliations:** 1Department of Life, Health and Environmental Sciences, School of Dentistry, University of L’Aquila, 67100 L’Aquila, Italy; rocco.franco@student.univaq.it (R.F.); ettore.lupi@graduate.univaq.it (E.L.); enzo.iacomino@graduate.univaq.it (E.I.); mario.capogreco@univaq.it (M.C.); maurizio.damario@univaq.it (M.D.); 2Dentistry Unit, Department of Pediatric Surgery, Bambino Gesù Children’s Hospital, IRCCS, Viale Ferdinando Baldelli 41, 00146 Rome, Italy; angela.galeotti@opbg.net; 3Institute of Endodontics, Faculty of Medicine, University of Coimbra, 3000-075 Coimbra, Portugal; 4Coimbra Institute for Clinical and Biomedical Research (iCBR) and Center of Investigation on Environment Genetics and Oncobiology (CIMAGO), Faculty of Medicine and Clinical Academic Center of Coimbra (CACC), 3000-548 Coimbra, Portugal

**Keywords:** stem cell transplantation, oral mucositis, low-level light therapy

## Abstract

Oral mucositis is a common and debilitating side effect induced by stem cell transplantation that is experienced by cancer patients undergoing chemotherapy or radiation therapy. This condition involves inflammation and ulceration of the oral mucosa, leading to pain, difficulty with eating and speaking, and an increased risk of infections. Mucositis not only compromises the quality of life for cancer patients, but also affects treatment outcomes and may necessitate dose reductions or treatment delays. This scientific article provides a comprehensive overview of mucositis. The purpose of this literature review with a meta-analysis is to evaluate the efficacy of laser therapy in treating post-transplant mucositis. *Materials and methods*: A search of the literature from 3 May 2023 was carried out on three online databases, PubMed, Scopus, and Web of Science. Only studies that treated patients with laser therapy were considered; only studies with the placebo-treated control group were considered. Review Manager version 5.2.8 (Cochrane Collaboration) was used for the pooled analysis. We measured the std. mean difference between the two groups (laser and placebo). *Results*: There were 230 papers included in this review. Two hundred twenty-seven were excluded. Furthermore, a manual search was performed. After the search phase, three articles were considered in the study. The overall effect showed differences in the degree of mucositis in the laser-treated patients compared with the placebo group. The meta-analysis shows a reduction in the degree of mucositis in the patients treated with laser therapy (std. mean difference −1.34 [−1.98; −0.98]; C.I. 95%). *Conclusions*: The application of laser therapy results in decreased severity of oral mucositis from radiation and chemotherapy. Our study shows that the application of low-level laser therapy in the treatment of transplant mucositis has excellent efficacy in relieving the symptoms and severity of mucositis.

## 1. Introduction

Cancer treatments like chemotherapy and radiation therapy have significantly improved patient survival rates. However, these treatments often have adverse effects, and oral mucositis is one of the most common and distressing complications [1]. Oral mucositis is one of the most common and upsetting side effects seen in cancer patients receiving therapy. Mucositis typically begins without symptoms, but can progress to cause redness, burning sensations, and an increased sensitivity to hot and spicy foods. In more severe cases, areas of skin may peel off and ulcers may form, leading to difficulty with swallowing and a reduced oral intake. These complications can greatly impact a patient’s quality of life [2]. Nearly all head and neck cancer patients who receive radiation also develop mucositis, as do about 20–40% of patients receiving conventional chemotherapy and 60–85% of patients receiving hematopoietic stem cell transplantation (HSCT). Guidelines for preventing oral mucositis have recently been released by the Multinational Association of Supportive Care in Cancer (MASCC) and the International Society of Oral Oncology (ISOO). Low-level laser therapy (LLLT), also known as photobiomodulation, was a suggested treatment for patients receiving head and neck radiotherapy [3]. Several papers discuss the impact of mucositis on cancer patients and explore the mechanisms contributing to its development [4]. Treatment- or patient-related risk factors can contribute to the onset of oral mucositis. HSCT conditioning regimens; induction therapy in leukemia patients; and hematologic malignancies are all treatment-related risk factors for mucositis in young patients. Other risk factors are tobacco use, patients’ gender, poor oral hygiene, etc. [5]. In fact, there has always been the question of how to prevent mucositis in patients receiving induction and maintenance chemotherapy for HSCT. The pathogenesis of oral mucositis is a multifactorial process involving various cellular and molecular events. It begins with direct cytotoxic damage to the oral mucosal cells by anticancer therapies, leading to inflammation and oxidative stress. Subsequently, pro-inflammatory cytokines, such as tumor necrosis factor alpha (TNF-α) and interleukin-1 beta (IL-1β), initiate a cascade of events that disrupt the normal regenerative processes of the oral mucosa, resulting in mucosal ulceration [6,7]. Oral mucositis worsens a patient’s quality of life [8]. The pain and discomfort associated with mucositis can lead to decreased oral intake, malnutrition, weight loss, and impaired speech. Additionally, mucositis increases the risk of local and systemic infections, often necessitating antibiotics and hospitalization [9,10]. The most frequent late and persistent adverse effect of radiation therapy for the head and neck is xerostomia, or dry mouth, which has a major negative influence on patients’ quality of life. The parotid glands are frequently exposed to radiation. Parotid dysfunction begins at levels of 10–15 Gy and can result in a 75% reduction in salivation at levels of 40–50 Gy. Xerostomia combined with mucositis can cause a nutrient deficit, weakening the patient even more [5]. The severity of mucositis can also result in treatment interruptions, dose reductions, or treatment discontinuation, compromising the efficacy of anticancer therapies. The management of oral mucositis involves a multidisciplinary approach aiming to prevent, alleviate, and treat symptoms [11,12,13,14]. Various interventions have been explored, including oral care protocols, cryotherapy, mucosal protectants, pain management strategies, and growth factors. Supportive care measures, such as maintaining good oral hygiene, avoiding irritants, and providing nutritional support, are crucial in preventing and managing mucositis [15,16,17]. Ongoing research focuses on identifying novel therapeutic targets and interventions to mitigate the development and severity of oral mucositis. These include molecularly targeted therapies, growth factors, anti-inflammatory agents, and cryotherapy techniques. Previous studies with meta-analysis have either evaluated laser treatment in pediatric patients or considered studies that used laser therapy either as a means of prevention or for the treatment of mucositis indiscriminately. In addition, other studies do not evaluate the need to distinguish the type of neoplasm in the sample considered or do not have a control group [2,18,19,20]. Advancements in personalized medicine and targeted therapies may offer new avenues for managing mucositis, minimizing treatment disruptions, and improving patient outcomes [21,22,23]. We felt the need to conduct this meta-analysis to help clinicians prevent mucositis due to HSCT because the topic is very important. Therefore, the purpose of this systematic review with meta-analysis is to evaluate the efficacy of laser therapy in treating mucositis resulting from HSCT in adult and pediatric populations.

## 2. Materials and Methods

### 2.1. Eligibility Criteria

We applied the following Population, Intervention, Comparator, and Outcomes (PICO) model to assess the document eligibility [24]:(P) Patients with cancer and oral mucositis developed after radiation therapy;(I) Treated with laser therapy;(C) Compared with patients treated with a placebo;(O) To assess the effectiveness of laser therapy in the treatment of mucositis.

As per the inclusion criteria, only randomized clinical trials providing data about the prevalence of oral mucositis according to the WHO Oral Mucositis Grading Objective Scale (or equivalent) in both groups were included. To ensure the accuracy of our study, we established specific exclusion criteria. These included (1) having an autoimmune disease; (2) undergoing a non-myeloablative conditioning regimen; (3) receiving haploidentical or autologous hematopoietic stem cell transplantation; (4) being part of a cross-over study design; (5) not having studies available in English; (6) only having access to posters and conference abstracts instead of full-text studies; (7) studying animals instead of humans; (8) including reviews rather than original study articles (topical or systematic); (9) case reports/series.

### 2.2. Search Strategy

To gather information for our research, we conducted a thorough search of PubMed, Web of Science, and Scopus databases for articles published up to 1 May 2023. Our search strategy is outlined in Table 1.

Additionally, we manually searched published systematic and topical reviews on similar topics. During this systematic review, we adhered to the Preferred Reporting Items for Systematic Reviews (PRISMA) 2020 criteria and the Cochrane Handbook for Systematic Reviews of Interventions. The International Prospective Register of Systematic Reviews (PROSPERO) has assigned the number 423968 to our systematic review procedure.

### 2.3. Extracting Data

The process of data extraction involved a manual review of each source to select relevant information. Two reviewers (RF and MDA) evaluated the extracted data independently, and any disagreements were resolved through a third reviewer (MC). The data extracted included the first author, year of publication, nationality, number and age of study participants, diagnostic criteria/tools used for mucositis diagnosis, correlation of mucositis grade between laser and placebo, and the study’s significance. All extracted data were recorded on a Microsoft Excel sheet.

### 2.4. Data Screening

The data were extracted after reading the articles and were processed with software and entered into an excel table so that they could then be processed for meta-analysis. The data were processed by 2 independent researchers.

### 2.5. Quality Assessment

The quality of the included papers was assessed by two reviewers, RF and EI, using the reputable Cochrane risk-of-bias assessment for randomized trials (RoB 2). The following six areas of possible bias are evaluated by this tool: random sequence generation, allocation concealment, participant and staff blinding, outcome assessment blinding, inadequate outcome data, and selective reporting. A third reviewer (MC) was consulted in the event of a disagreement until an agreement was reached.

### 2.6. Statistical Analysis

The software Review Manager version 5.2.8 (Cochrane Collaboration, Copenhagen, Denmark; 2014) was used to perform the pooled analysis. We measured the risk ratio (RR) between the two groups (laser therapy and placebo). The Higgins Index (*I2*) and the chi-squared test were implemented to assess Heterogeneity among studies. We classified heterogeneity as follows: low heterogeneity (<30%), medium heterogeneity (30–60%), and high heterogeneity (>60%).

## 3. Results

### 3.1. Study Characteristics

According to Figure 1, three studies were examined in the meta-analysis. A total of 230 articles were selected because of the search. A total of 73 papers were excluded before the screening; 6 articles were not in English, and 67 were duplicates. The remaining 157 articles were selected for the title and abstract screening to evaluate whether they met the PICO criteria. One hundred forty-two articles were excluded because only RCTs were selected; therefore, fifteen articles were selected. Among these, ten were excluded because they do not respond to the PICO questions, and two were off topic. Therefore, the remaining three articles were selected for the meta-analysis. The included studies were published between 2008 and 2017. The three included studies were randomized controlled trials. All of these studies showed the difference in the severity of mucositis in patients receiving laser therapy and placebo. The data extracted from each study are reported in Table 2.

For more information, visit: http://www.prisma-statement.org/ (accessed on 1 March 2023).

### 3.2. Main Findings

There are 98 included subjects in this review. The average age of the study participants is 17.7 years old. The WHO scale evaluated all the patients.

The meta-analysis values were compared seven days after laser therapy to have more data homogeneity. The World Health Organization (WHO) has classified mucositis into the following five grades: Grade 0, indicating no mucositis; Grade I, indicating erythema without lesions; Grade II, indicating the presence of ulcers but the ability to feed; Grade III, indicating painful ulcers but still able to consume liquid food with the help of analgesics; and Grade IV, indicating the need for enteral or parenteral feeding and continuous analgesia.

The Vitale study [25] evaluated pediatric patients suffering from oral mucositis undergoing chemotherapy and hematopoietic stem cell transplantation. The patients were randomly divided into a study group to be treated with laser therapy and a placebo group. There were 16 patients enrolled. Eight patients were placed in the laser therapy group and eight were placed in the control group. The WHO Oral Mucositis Grading Objective Scale was used to assess mucositis and the Visual

Analogue Scale (VAS) was used to assess discomfort. The patients were observed and evaluated three, seven, and eleven days following the first day of laser therapy. The patients in the laser group were treated with HSCT for four consecutive days, once a day, and the placebo group underwent sham treatment. The VAS scale evaluated all patients, and then the degree of mucositis was assessed at 0, 3, 7, and 11 days. Statistical examinations via Analysis of Variance (ANOVA) and chi-squared test were performed to assess correlation over time. All patients in the laser group showed a decrease in VAS, from a median of 8.25 at day 0 to 4.75 after three days, 2.75 after seven days, and 1.25 after eleven days. The patients in the placebo group experienced a decline in VAS, with a median of 7.5 at day 0, falling to 5 after three days, 3.38 after seven days, and 2.25 after eleven days. A statistically significant decline in VAS was already apparent in the laser group by day 3 (*p* < 0.05) and in the sham group by day 7 (*p* < 0.05). The patients in the laser group had a mucositis regression from a median of 3.38 at day 0 to 2.75 after three days. In contrast, patients in the sham group had a regression from a median of 3.38 at day 0 to a median of 3.38 after three days, and from a median of 2.38 after seven days to a median of 1.88 after eleven days. On day 7, solely in the laser group, there was a statistically significant regression of OM (*p <* 0.05) [25].

Kuhn’s study evaluated the effects of laser therapy as an adjuvant for reducing oral mucositis in pediatric patients undergoing HSCT. This randomized trial looked at patients from 2005 to 2006 and divided them into study and control groups. There were 21 patients recruited, divided into 9 patients in the study group and 11 in the placebo group. The patients were treated with laser applications every day until the complete regression of symptoms. Since diagnosis, the patients were monitored daily on the evolution of mucositis. The patients were assessed with the WHO scale. In addition, a multiple linear regression was performed to evaluate the association between the two groups of patients. After treatment, OM was discovered, on average, 6.6 days (range: 5.0 to 7.5) later. In the laser group and the sham group, the median of OM in grades on the first day of diagnosis was 3 (2–4), and the means were 3.1 (2–4) and 3.4 (2–4), respectively (*p* = 0.82). No patients had OM grade 1 symptoms. Over five days, all patients received laser treatment (group A) or a placebo (group B). The applications of lasers were well tolerated, and their use had no adverse side effects. The most often affected areas were the lateral/ventral tongue (40%) and the floor of the mouth (65%). The OM grade gradually decreased, and all patients’ lesions completely disappeared. In the laser group and the sham group, respectively, 1 in 9 and 9 in 12 patients exhibited OM (grade 2 or more) on day seven after OM diagnosis (*p* = 0.029) [26].

Khouri’s study evaluated a group of 22 patients with chemotherapy and stem cell transplantation. A total of 12 patients were treated with laser therapy. At the time of symptom onset and for the next 15 days, they were randomly treated with laser therapy or with mouthwash in the control group. The evolution of mucositis was evaluated over time. A statistical analysis via the Wilcoxon test was performed to evaluate the correlation between the change in mucositis between the two groups. When the OM progression in the patients from groups I and II was compared, it became clear that group I’s OM frequency was lower, and the difference between the groups was statistically significant (*p* = 0.02). Group I showed a mean mucositis grade of 1.75 0.45, while group II showed a mean of 2.45 0.93. According to the WHO scale, there was a statistically significant difference between the groups (*p* < 0.01) [27].

### 3.3. Meta-Analysis

The included studies had medium heterogeneity (*I*2 = 38%). Therefore, the meta-analysis was conducted using the random model effect. We considered the degree of mucositis with laser and placebo treatment as the outcome. During the meta-analysis, we used the WHO index as a standard method to assess the degree of mucositis. Also, given the lack of studies, we did not consider both the type of cancer they have and their age.

The overall effect, reported in the forest plot (Figure 2), showed that there was a difference in the mucositis degree between the laser and placebo groups (std. mean difference −1.34 *[*−*1.98;* −*0.98]*; C.I. 95%), suggesting that a laser is a beneficial tool for the treatment of mucositis.

### 3.4. Quality Assessment and Risk of Bias

The risk of bias in the included studies is reported in Figure 3. Regarding the randomization process, one study presents a high risk of bias and allocation concealment. All other studies ensure a low risk of bias. Only one study excludes a performance; two studies confirm an increased risk of detection bias (self-reported outcome), and two of the included studies present a low detection bias (objective measures) (Figure 3). Two studies ensure a low risk regarding attrition and reporting bias.

## 4. Discussion

OM is the most frequent and unpleasant side effect experienced by patients receiving large doses of chemotherapy, radiation, or HSCT [28,29]. Clinical effects include pain and challenges with speaking, swallowing, and eating. The most recent therapy for preventing and treating OM is using palifermin, a recombinant human form of the epithelial cell stimulant keratinocyte growth factor (KGF). Palifermin was recently investigated as one of several medicines for preventing and treating OM [30]. Significant improvements in pain management and a decrease in the occurrence and duration of severe OM were among the outcomes. According to Rubenstein et al., some therapeutic drugs, such as chlorhexidine (prevention), amifostine (treatment), and chamomile (prevention and treatment), have insufficient evidence in the literature on the prevention and treatment of mucositis [31].

The main auxiliary drugs used for the treatment of mucositis are anti-inflammatory drugs, including the use of antimicrobials such as chlorhexidine. However, each of these drugs has side effects. In fact, laser therapy is free of side effects.

According to the present systematic review, low-level laser therapy (LLLT) can be a valuable tool for reducing mucositis in individuals receiving chemotherapy.

The objective of Vitale’s clinical trial was to assess the effectiveness of laser therapy in oncohematological patients with OM [25]. Mucositis is undoubtedly one of the most disabling side effects of chemotherapy, and it affects children and adolescents with ALL or lymphomas more frequently. In oncohematological patients receiving high-dose chemotherapy, mucositis often starts 2–12 days after chemotherapy and can become quite bad 7–14 days after treatment starts. Oral lesions typically last for two to three weeks, but they may last longer in patients with severe neutropenia. Cytokines mediate five stages of pathogenesis in OM [31,32]. In the first stage, antineoplastic drugs directly damage cellular DNA. Afterward, there is an increase in transcription factors and signal production, as well as a decrease in cellular turnover. This leads to an increase in apoptosis and tissue damage. Biological changes during this phase can produce excruciating pain despite the tissue’s integrity. This clarifies that the pain being referred to may not always be related to the visual appearance of the lesions. Consequently, bacterial or fungal superinfections in the ulcerative stage 7–10 days following treatment are highly probable.

Angiogenesis and cell proliferation are features of the healing period. In the literature, there is a debate over the effectiveness of LLLT in lowering the severity of OM in young HSCT patients. In a systematic review, Qutob et al. pointed out that there are conflicting results when it comes to using LLLT to prevent OM in children [33,34]. The use of high-power laser therapy (HPLT) to treat inflammatory disorders has demonstrated that lesions heal more quickly after treatment. In the current study, all patients reported difficulty with oral eating and moderate to severe pain due to ulcerations and erythema. After the first laser application, all patients in the laser group experienced a statistically significant reduction in pain, and on day 11, complete healing and pain regression was achieved. According to the research, laser therapy’s anti-inflammatory and analgesic effects significantly improve all lesions and pain perception. Numerous branches of medicine and dentistry have investigated laser therapy. Comparing the results of laser light trials for preventing or curing OM has been challenging due to the lack of protocol uniformity and the wide variability in studied wavelengths. Additionally, research on the use of lasers for chemotherapy-induced OM has primarily focused on adult cancer patients.

Most of the studies evaluate the effect of laser therapy ton the onset of mucositis. In one study, Barasch et al. used laser therapy as a preventive measure in 20 cancer patients. They administered laser treatment on either the right or left side of the midline, with the opposite side serving as a control [32]. The therapy began the day after chemotherapy and continued for five consecutive days. A randomized multicenter experiment conducted by Bensadoun et al. found that patients who received laser treatment reported experiencing less pain and oral mucositis (OM) (*p <* 0.05) compared to the control group [35].

Although the studies indicate that using laser therapy as a preventive measure may lessen the severity of OM, the impact on pain intensity and swallowing ability is somewhat debatable. There is considerable interest in researching the role of laser therapy in establishing chemotherapy-induced OM because the laser can reduce pain and speed up the healing process in non-cancer and cancer patients. In comparison to the sham group, patients who received infrared laser treatment had significantly less pain after 7 days (*p* = 0.008) and 15 days (*p* = 0.0009). However, this study’s drawbacks are that the patient group assignment was not randomized, and the applications were submitted 48 h apart. The Kuhn trial is the initial randomized placebo-controlled investigation to determine if LLLT helps to hasten wound healing in kids with chemotherapy-induced OM [36,37,38].

AlGaInP (660 nm) and GaAlAs (780 nm) lasers with a 25 mW power and 6.3 J/cm^2^ dosage were used to irradiate group I in this work. The two lasers were alternately employed from the beginning of the conditioning program until the D + 15 post transplantation. According to Antunes et al., OM was evaluated using the WHO and Oral Mucositis Assessment Scale (OMAS) measures [39].

Studies have shown that therapeutic lasers can effectively prevent and treat OM. The results are similar to those obtained by Antunes et al. Additionally, the use of therapeutic lasers has not caused any side effects or discomfort. However, more research is needed to confirm these findings, especially with a larger group of patients and standardized irradiation techniques for those receiving high doses of chemotherapy. Overall, therapeutic lasers have demonstrated promising results in the prevention and treatment of OM.

One limitation of this review is the small number of studies that meet the inclusion criteria, and another concerns the age of the patients. Two studies were performed on pediatric patients, while one was performed on young adult patients.

## 5. Conclusions

Oral mucositis remains a significant challenge in the management of cancer patients. The pathogenesis of mucositis is complex, involving multiple cellular and molecular factors. While several management strategies exist, further research is necessary to improve prevention, early intervention, and treatment options. By addressing the impact of mucositis on cancer patients and exploring novel therapeutic approaches, healthcare providers can strive to enhance patients’ quality of life during cancer treatment and optimize treatment outcomes. Therefore, it will be interesting to evaluate further studies on the role of laser therapy in preventing mucositis. From the results of this meta-analysis, we can state, despite the limitation of the sample, that laser therapy is effective in treating mucositis post-transplantation. Given the importance of the topic, our study with meta-analysis is proposed as a guide for clinicians involved in specialty dentistry since the use of laser therapy for the treatment of mucositis allows for patients to have regular nutrition and a better quality of life. Few studies were considered, and the various types of lasers and powers were not taken into account. More clinical trials are needed to be able to evaluate the efficacy of laser therapy in patients undergoing chemotherapy for HSCT. In fact, this meta-analysis of ours has an exiguous sample of patients due to the paucity of studies, and we did not distinguish the type of laser used and the power. However, given the clinical importance of its use, we felt that this review would provide clarity and help clinicians improve the quality of life of these patients.

## Figures and Tables

**Figure 1 medicina-59-01413-f001:**
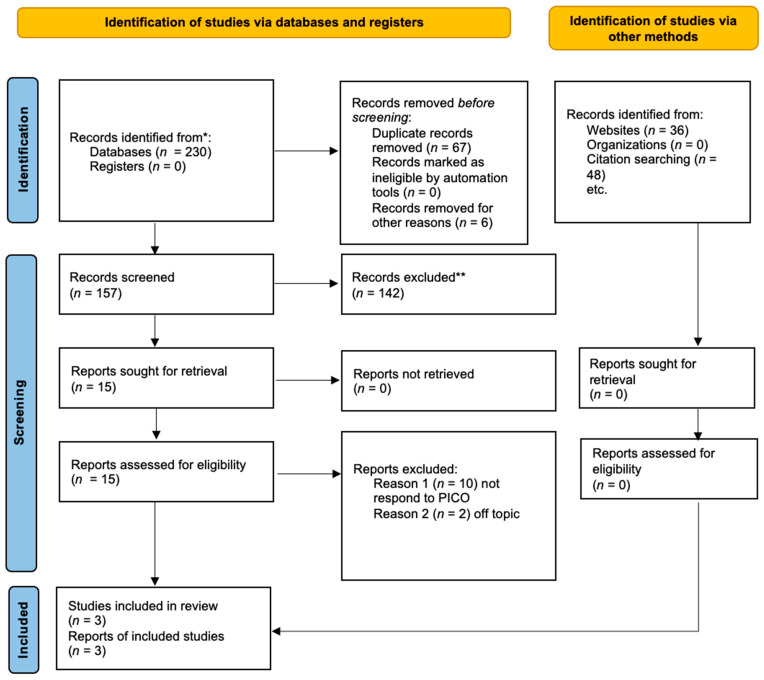
PRISMA flowchart based on the following: Page, M.J.; McKenzie, J.E.; Bossuyt, P.M.; Boutron, I.; Hoffmann, T.C.; Mulrow, C.D.; et al. The PRISMA 2020 statement: An updated guideline for reporting systematic reviews [24]. * studies identified by search methods; ** studies removed because they are systematic reviews of the literature.

**Figure 2 medicina-59-01413-f002:**
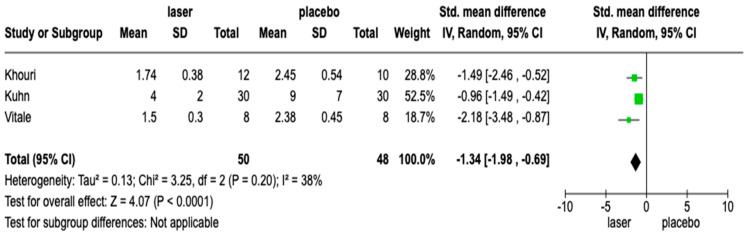
Forest plot of studies included in the meta-analysis; C.I. 95%.

**Figure 3 medicina-59-01413-f003:**
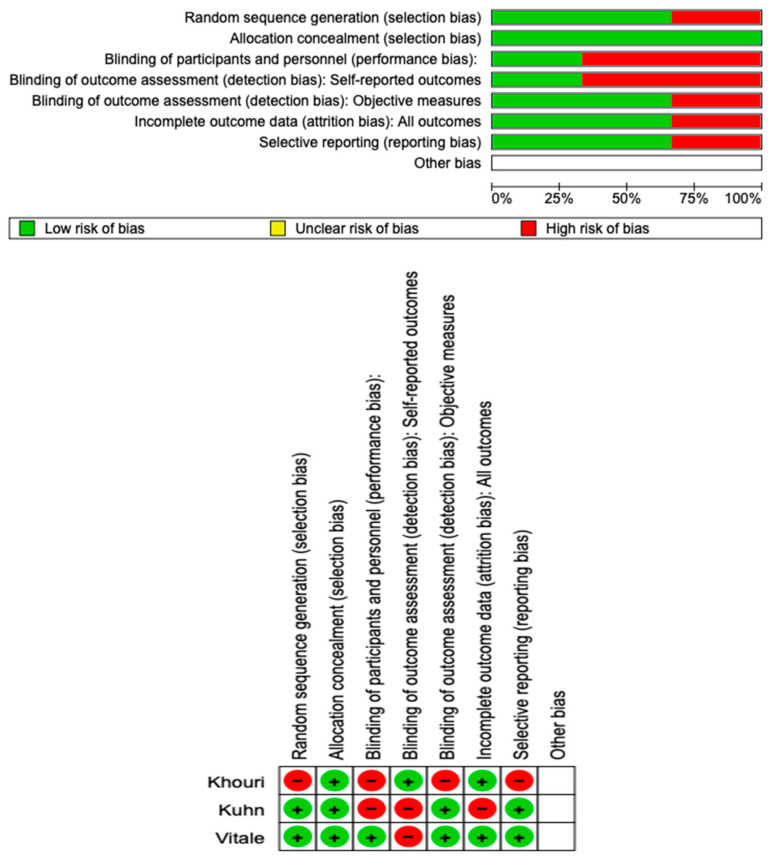
Risk of bias; red indicates high risk, and green indicates low risk of bias.

**Table 1 medicina-59-01413-t001:** Search strategy.

**PubMed** Search: stem cell transplantation AND laser AND oral mucositis (“stem cell transplantation” [MeSH Terms] OR (“stem” [All Fields] AND “cell” [All Fields] AND “transplantation” [All Fields]) OR “stem cell transplantation” [All Fields]) AND (“laser s” [All Fields] OR “lasers” [MeSH Terms] OR “lasers” [All Fields] OR “laser” [All Fields] OR “lasered” [All Fields] OR “lasering” [All Fields]) AND (“stomatitis” [MeSH Terms] OR “stomatitis” [All Fields] OR (“oral” [All Fields] AND “mucositis” [All Fields]) OR “oral mucositis” [All Fields])
**Scopus** TITLE-ABS-KEY (stem cell transplantation AND laser AND oral mucositis)
**Web of Science** stem cell transplantation [all field] and laser [all field], and oral mucositis [all field]

**Table 2 medicina-59-01413-t002:** Principal elements of the studies that formed part of the present systematic analysis. * Statistically significant correlation.

Author	Year	Nationality	Number of Case vs. Control	Age	Diagnostic Tool of Mucositis	Correlation of Mucositis Grade between Laser and Placebo	Significance of Study
Vitale [25]	2017	Italy	16 patients: 8 laser group 8 placebo	12.4 yrs	WHO	Vas e WHO Laser Group: T0 3.38 T1 3 days 2.73 T2 7 days 1.50 T3 11 days 0.38 Placebo: T0 3.38 T1 3 days 3.38 T2 7 days 2.38 T3 11 days 1.88 (*p <* 0.05) *	Laser is an excellent aid for the treatment of mucositis
Kuhn [26]	2008	Brazil	21 patients 9 laser group 12 placebo	8.2 yrs	WHO	Laser Group: T0 3.0 T1 3 days 2.3 T2 7 days 0.73 T3 11 days 0 Placebo: T0 3.1 T1 3 days 2.8 T2 7 days 1.8 T3 11 days 0.54 *p* = 0.0029 *	Laser is an excellent aid for the treatment of mucositis
Khouri [27]	2009	Brazil	22 patients: 12 laser 10 control	32.7 yrs	WHO	Laser Group: T0:3.0 T1 7 days 1.74 Placebo: T0 3.0 T2 7 days 2.45 *p* = 0.001 *	Laser is an excellent aid for the treatment of mucositis

## Data Availability

Data will be made available upon request to the corresponding author.

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
