# Peer review of "Low-Level Laser Therapy for the Treatment of Oral Mucositis Induced by Hematopoietic Stem Cell Transplantation: A Systematic Review with Meta-Analysis"

_medicina, 2023, doi:10.3390/medicina59081413_

Round 1

Reviewer 1 Report

The authors submitted a systematic review with meta-analysis aimed at evaluating the efficacy of laser therapy in treating mucositis resulting from radiation and chemotherapy. As one of the most common adverse effects of cancer treatments is oral mucositis, it is crucial to assess the efficacy of the different treatment options in enhancing the quality of life of these patients. The review itself is well conducted and well structured. The PRISMA protocol is the major strength of the overall paper. 

Overall, the Authors may find useful the following suggestions in order to improve the quality of their manuscript.

1. Please, be sure to use only keywords accordingly to medical subject headings (Mesh word) for a better indexing.

2. Lines 65 and 77: The Authors may find useful the following papers https://doi.org/10.3390/dj7020060 and https://doi.org/10.1007/978-3-319-65421-8_11

3. I suggest the Authors to move Figure 1 before Table 2 in order to assist the reader. 

4. Please, better describe the limitations of your study. 

Minor editing of English language is required. Please, carefully check and revise English usage and style through the text.

Author Response

The authors submitted a systematic review with meta-analysis aimed at evaluating the efficacy of laser therapy in treating mucositis resulting from radiation and chemotherapy. As one of the most common adverse effects of cancer treatments is oral mucositis, it is crucial to assess the efficacy of the different treatment options in enhancing the quality of life of these patients. The review itself is well conducted and well structured. The PRISMA protocol is the major strength of the overall paper. 

Answer: Thank you for your positive feed-back and criticism to help us improve our paper.

Overall, the Authors may find useful the following suggestions in order to improve the quality of their manuscript.

  1. Please, be sure to use only keywords accordingly to medical subject headings (Mesh word) for a better indexing.

Answer: Keywords were changed, as suggested “stem cell transplantation; oral mucositis; low-level light therapy

  1. Lines 65 and 77: The Authors may find useful the following papers https://doi.org/10.3390/dj7020060

 and https://doi.org/10.1007/978-3-319-65421-8_11

Answer: One paper is related to anorexia and we did not find relevance for this review but we included the second as suggest “The most frequent late and persistent adverse effect of radiation therapy for the head and neck is xerostomia, or dry mouth, which has a major negative influence on patients' quality of life. The parotid glands are frequently exposed to radiation. Parotid dysfunction begins at levels of 10-15 Gy and can result in a 75% reduction in salivation at levels of 40–50 Gy. Xerostomia, combined with mucositis can cause a nutrient deficit weakening the patient even more”.

  1. I suggest the Authors to move Figure 1 before Table 2 in order to assist the reader. 

Answer: Thank you for your suggestion, it has been changed accordingly.

  1. Please, better describe the limitations of your study. 

Answer: “More clinical trials will be needed to be able to evaluate the efficacy of laser therapy in patients undergoing chemotherapy for HSCT. In fact, this meta-analysis of ours has an exiguous sample of patients, due to the paucity of studies, and we did not distinguish the type of laser used and the power. However, given the clinical importance of its use anyway, we felt it would provide clarity and help clinicians improve the quality of life of these patients.“

Thank you for the advice and for reviewing our paper.

Reviewer 2 Report

Dear authors,

I would like to provide the following comments:

Why is literature review used in the title instead of systematic review?

The referencing of the introduction should be corrected. Referencing at the end of a long paragraph does not seem correct and it is better to be a little more detailed.

A similar study has been conducted by Mengxue He et al (ttps://doi.org/10.1007/s00431-017-3043-4), Please mention in the introduction the necessity of conducting a new systematic review by referring to the differences between the present study and the mentioned study.  Previous systematic review was more comprehensive since it included eight primary study and only one study was added differently here, while the outcomes considered in the previous systematic review were more (prevention or treatment of oral mucositis, e.g., incidence, severity, duration in days, and pain intensity).

Please refer to the articles that introduced the PRISMA checklist in the method section (line 120). Also, if it is possible refer to “Cochrane Handbook for Systematic Reviews of Interventions”.

Author Response

Dear authors, 

I would like to provide the following comments:

Answer: Thank you for your positive feed-back and criticism to help us improve our paper.

Why is literature review used in the title instead of systematic review?

Answer: Title as been changed as suggested: “Low laser therapy for the treatment of oral mucositis induced by hematopoietic stem cell transplantation: systematic review with meta-analysis”.

The referencing of the introduction should be corrected. Referencing at the end of a long paragraph does not seem correct and it is better to be a little more detailed.

Answer: We agree with your suggestion and made modifications accordingly.

A similar study has been conducted by Mengxue He et al (ttps://doi.org/10.1007/s00431-017-3043-4), Please mention in the introduction the necessity of conducting a new systematic review by referring to the differences between the present study and the mentioned study.  Previous systematic review was more comprehensive since it included eight primary study and only one study was added differently here, while the outcomes considered in the previous systematic review were more (prevention or treatment of oral mucositis, e.g., incidence, severity, duration in days, and pain intensity).

Answer: The following rationale has been introduced: “Previous studies with meta-analysis have either evaluated laser treatment in pediatric patients, or considered studies that used laser either as a means of prevention or for treatment of mucositis indiscriminately. In addition, other studies have not evaluated the need to distinguish the type of neoplasm in the sample considered or don’t have control group”

Please refer to the articles that introduced the PRISMA checklist in the method section (line 120). Also, if it is possible refer to “Cochrane Handbook for Systematic Reviews of Interventions”.

We have made these changes. I thank you for the valuable advice.

Reviewer 3 Report

This study focused on oral mucositis as a common and debilitating side effect experienced by cancer patients undergoing chemotherapy or radiation therapy induced by stem cell transplantation. The work is a significant contribution to the field. However, some comments can be help in its improving.

The abstract needs to improve. Please improve the conclusion of abstract.

The introduction suffer from the lack of enough references.

Please mention the data screening.

Please improve the quality of images.

Conclusion should be improve. It dose not support the results.

Generally, the manuscript is written well and can be accepted after these minor corrections.

Minor editing of English language required.

Author Response

This study focused on oral mucositis as a common and debilitating side effect experienced by cancer patients undergoing chemotherapy or radiation therapy induced by stem cell transplantation. The work is a significant contribution to the field. However, some comments can be help in its improving. 

Answer: Thank you for your positive feed-back and criticism to help us improve our paper.

The abstract needs to improve. Please improve the conclusion of abstract.

Answer: The abstract has been changed. “… Our study showed that the application of low laser therapy in the treatment of transplant mucositis has excellent efficacy in relieving the symptoms and severity of mucositis.

The introduction suffer from the lack of enough references.

Answer: New references were added.

Please mention the data screening.

Answer: ”The data were extracted after reading the articles and were processed with software and entered into an excel table so that they could then be processed for meta-analysis. The data were processed by 2 independent researchers”.

Please improve the quality of images.

Conclusion should be improve. It dose not support the results.

Answer – Conclusion has been modified : ”Oral mucositis remains a significant challenge in the management of cancer patients. The pathogenesis of mucositis is complex, involving multiple cellular and molecular factors. While several management strategies exist, further research is necessary to improve prevention, early intervention, and treatment options. By addressing the impact of mucositis on cancer patients and exploring novel therapeutic approaches, healthcare providers can strive to enhance patients' quality of life during cancer treatment and optimize treatment outcomes. From the results of this meta-analysis, we can state, albeit the limitations inherent to the low overall sample size of patients, that low laser therapy is effective in treating mucositis post-transplantation, reducing the average severity of OM and oral pain. Therefore, this treatment improves patient’s quality of life. More clinical trials will be needed to allow the assessment of the efficacy of laser therapy in patients undergoing chemotherapy for HSCT, to better define which type of lasers, power, frequency and time of exposure will allow optimized outcomes.”

Generally, the manuscript is written well and can be accepted after these minor corrections.

Thank you for reviewing our manuscript and giving your valuable advice.

Round 2

Reviewer 2 Report

Thanks to the authors for the corrections.

Author Response

Thank you for the invaluable cooperation.